# Rituximab Therapy for Adults with Nephrotic Syndromes: Standard Schedules or B Cell-Targeted Therapy?

**DOI:** 10.3390/jcm10245847

**Published:** 2021-12-13

**Authors:** Lucia Del Vecchio, Marco Allinovi, Paolo Rocco, Bruno Brando

**Affiliations:** 1Department of Nephrology and Dialysis, Sant’Anna Hospital, ASST Lariana, 22042 Como, Italy; 2Nephrology, Dialysis and Transplantation Unit, Careggi University Hospital, 50134 Florence, Italy; marco.allinovi@gmail.com; 3Department of Pharmaceutical Sciences, Università degli Studi di Milano, Via G. Colombo, 71-20133 Milan, Italy; paolo.rocco@unimi.it; 4Haematology Laboratory and Transfusion Centre, Legnano General Hospital (Milan), 20025 Milan, Italy; bruno.brando@asst-ovestmi.it

**Keywords:** rituximab, nephrotic syndrome, membranous nephropathy, B lymphocyte, CD19, focal segmental glomerulosclerosis, high-resolution flow cytometry

## Abstract

Rituximab is a chimeric anti-CD20 monoclonal antibody. It acts mainly through complement-dependent cytotoxicity on B cells expressing the CD20 marker. In this review, we analyse the efficacy and possible pitfalls of rituximab to treat nephrotic syndromes by taking into account pharmacological considerations and CD19 marker testing utility. Despite the fact that the drug has been in use for years, efficacy and treatment schedules in adults with nephrotic syndrome are still a matter of debate. Clinical trials have proven the efficacy and safety of rituximab in idiopathic membranous nephropathy. Data from observational studies also showed the efficacy of rituximab in minimal change disease and focal segmental glomerulosclerosis. Rituximab use is now widely recommended by new Kidney Disease Improved Outcome (KDIGO) guidelines in membranous nephropathy and in frequent-relapsing, steroid-dependent minimal change disease or focal segmental glomerulosclerosis. However, rituximab response has a large interindividual variability. One reason could be that rituximab is lost in the urine at a higher extent in patients with nonselective nephrotic proteinuria, exposing patients to different rituximab plasma levels. Moreover, the association between CD19+ levels and clinical response or relapses is not always present, making the use of this marker in clinical practice complex. High resolution flow cytometry has increased the capability of detecting residual CD19+ B cells. Moreover, it can identify specific B-cell subsets (including IgG-switched memory B cells), which can repopulate at different rates. Its wider use could become a useful tool for better understanding reasons of rituximab failure or avoiding unnecessary retreatments.

## 1. Introduction

Starting from the mid-20th century, in parallel with the birth and development of modern nephrology, glomerulonephritis had received a systematic histopathological classification and the idea caught on that therapeutic choices should be based accordingly. In particular, the development of immunofluorescence techniques showed the presence of immunoglobulin and complement factors trapped inside the glomeruli, supporting the role of the immune system in the pathogenesis and use of immunosuppressive agents for their treatment [1]. In this context, humoral immunity and antibody-producing B lymphocytes are often involved. They also contribute to tissue injury by producing inflammatory cytokines and by presenting antigen to T lymphocytes [2].

In recent years, it has become clear that some glomerulonephritis are full-fledged autoimmune diseases. Accordingly, antibodies against the M-type phospholipase A2 receptor (PLA2R), a podocyte membrane glycoprotein, are detected in nearly 70 to 80% of patients with membranous nephropathy (MN) [3]; their titre is related to disease severity [4] and treatment response [5]. A small percentage of patients are found positive for antibodies targeting to another podocyte protein, the thrombospondin type-1 domain-containing 7A (THSD7A) [6]. More recently, other autoantigens were described [7,8], adding further complexity to the understanding of the pathogenesis of MN and to its distinction in primary or secondary forms [7].

The pathogenetic mechanisms at the basis of minimal change disease (MCD) and focal segmental glomerulosclerosis (FSGS) are much less clear. Both diseases affect the podocyte but with a different clinical presentation and treatment response, with MCD often transitioning to FSGS over its course (especially in those who develop treatment resistance) [9]. They probably represent a disease spectrum of different morphologic manifestations and stages and overlapping etiologic factors, including genetic ones, all causing nephrotic or subnephrotic syndrome as a consequence of podocyte damage [9]. Both humoral and cellular immunity have been implicated in their pathogenesis with non-well defined circulating factors damaging podocytes [10,11].

Despite significant improvement in the understanding of the pathogenesis, the treatment of nephrotic syndrome has remained quite aspecific, with steroids and immunosuppressants given with different doses and schedules representing the most used methods. These treatments are effective in many cases. However, the use of high-dose steroids and immunosuppressants can be burdened by several side effects and toxicity. Moreover, a significant number of patients are treatment resistant and often progress to end-stage kidney failure.

## 2. Rational and Mechanisms of Action of Rituximab

Rituximab is a chimeric monoclonal antibody (MoAbs) that depletes B cells expressing the CD20 antigen. Structurally, the drug is made by the fusion between the binding regions of the original murine anti-human CD20 with human IgG1 heavy chain and human kappa light-chain constant regions [12].

CD20 is a calcium-dependent co-stimulatory receptor of B cells, expressed on the cell membrane in the central phase of B lymphocyte development [13]. The rationale for using rituximab in B-cell targeted protocols resides in the particular kinetics of expression of this CD20 during B cell maturation. When CD20+ effectors and memory B lymphocytes are depleted, the activation chain leading to allo- and auto-antibody production is blocked. By the same token, early B cell precursors are spared, ensuring the survival of immature cells able to regenerate new antigen-naïve B cells, at least initially devoid of pathogenic autoreactivity. Mature plasmacells are also spared, thus preserving long-term defensive antibody memory [14]. Moreover, the high level of expression of CD20 on B cell membrane (around 150,000 antigen copies per B cell) and the closeness of its relevant epitopes to the B cell surface favour a high concentration of MoAb on its target and the optimal development of a series of cytotoxic interactions [15].

The main mechanism of action of rituximab on B cells is complement-dependent cytotoxicity mediated by the Fc portion of the antibody [16]. When rituximab binds two adjacent CD20 molecules on B cell surface, a complex molecular interaction takes place, greatly increasing the affinity for complement fixation [17]. This structural model is the key for subsequent complement recruitment. Complement activation may also have a causative role also in infusion toxicity during rituximab treatment [18].

A second mechanism of action of rituximab is antibody-dependent cellular cytotoxicity by a variety of effector cells, including natural killer (NK) cells, granulocytes and macrophages [16]. NK cells and presumably complement factors could have a synergistic effect [19]. This could be of interest from a clinical point of view, since both parameters can be deficient in various clinical settings.

Finally, studies have suggested that rituximab can act directly on podocytes to stabilize the cytoskeleton and improve proteinuria through an apparent B-lymphocyte independent mechanism [20,21]. The fact that rituximab effects go beyond CD20+ lymphocyte depletion could be a possible explanation why in clinical practice the correlation between the number of CD20+ cells and disease activity is not perfect and that in some patients the treatment effect seems to persist even after CD20 count normalisation.

## 3. Rituximab Therapy in Nephrotic Syndrome

New KDIGO guidelines on glomerulonephritis suggests a wider use of rituximab in adult patients with nephrotic syndrome than what was recommended eight years ago, reflecting consistent changes in available evidence and clinical practice [22].

The higher degree of evidence on rituximab use is available for MN, as several randomised clinical trials have become available in recent years. The Prospective Randomized Multicentric Open Label Study to Evaluate Rituximab Treatment for Idiopathic Membranous Nephropathy (GEMRITUX) trial [23] compared two 375 mg/m^2^ rituximab infusions one week apart with no therapy in 75 patients and found that a higher number of patients receiving rituximab achieved complete or partial remission of proteinuria at six months (albeit the difference was not statistically significant). This went together with a higher number of patients experiencing anti–PLA2R antibodies depletion. However, after a longer observation period (median follow-up was 17 months), remission occurred in 64.9% of the patients treated with rituximab compared to only 34.2% in those treated with supportive therapy. Safety was comparable in the two groups.

In the Membranous Nephropathy Trial Of Rituximab (MENTOR) trial [24], 130 subjects were assigned either to rituximab (1000 mg on days 1 and 15) or cyclosporine with a starting dose of 3.5 mg/kg/day then given for six to fourteen months depending on proteinuria response (tapered and discontinued over two months if remission was observed at six months).

In comparison to cyclosporine, rituximab obtained complete or partial remission in a higher percentage of patients at one year (52% vs. 62%, *p* = 0.004 for non-inferiority) and two years (20% vs. 60%, *p* < 0.001 for both non-inferiority and superiority). This went together with a better safety profile and a faster and larger decline in anti PLA2R titre (in those achieving remission) favouring rituximab. However, the big difference in remission at two years was partially influenced by proteinuria recurrence in some patients following cyclosporine discontinuation [25]. Moreover, the enrolment of patients with moderate CKD may have increased the number of side effects related to cyclosporine use [25].

Recently, a meta-analysis of 21 studies (603 patients) confirmed the efficacy of rituximab in MN, showing an overall remission rate of 67% and a mean proteinuria decrease of −4.90 g/day (95% CI −6.18, −3.63); rituximab efficacy was not significantly related to anti-PLA2R antibody status or previous immunosuppressive therapy [26].

The Sequential Treatment with Tacrolimus and Rituximab versus Alternating Corticosteroids and Cyclophosphamide in PMN study (STARMEN) compared treatment with corticosteroids and cyclophosphamide to a sequential regimen of tacrolimus (0.05 mg/Kg/day for six months then decreased and stopped at month nine) and rituximab (1 g at months six). The rational of the combination is that the non-immunosuppressive effects of calcineurin inhibitors could be reinforced by rituximab and that the latter could prevent relapses at the time of calcineurin inhibitor decalage or interruption. The combination of steroids and cyclophosphamide showed superiority in comparison to tacrolimus plus rituximab in achieving complete or partial remission at 24 months (83.7% vs. 58.1%, respectively; RR 1.44; 95% CI 1.08 to 1.92) [27]. The percentage of severe adverse events was comparable in the two groups (even if those randomized to the Ponticelli’s regimen experienced more severe infections, more adverse events and more adverse events per patient).

The Rituximab versus Steroids and Cyclophosphamide in the Treatment of Idiopathic Membranous Nephropathy (RI-CYCLO) trial [28] compared two rituximab infusions (1 g) two weeks apart to a regimen with corticosteroids and cyclophosphamide. At 24 months, the probability of complete or partial remission was comparable in rituximab and cyclic regimen (85% vs. 81%, respectively). The reduction of anti-PLA2R1 serum levels during follow-up occurred more rapidly in the rituximab arm and was accompanied by a decrease in proteinuria. The percentage of severe adverse events was comparable in the two groups (even if the patients randomized to rituximab experienced more drug infusion reactions).

Table 1 summarises the main efficacy and safety data of the GEMRITUX, MENTOR, STARMEN, and RI-CYCLO trials.

Recently, another study compared low-dose rituximab (a single dose of 375 mg/m^2^), a standard rituximab protocol (four weekly doses of 375 mg/m^2^) with Ponticelli’s regimen (cyclic regimen with corticosteroids and cyclophosphamide) [29], obtaining similar data to those of the RI-CYCLO study [28].

According to the 2021 KIDIGO guidelines [22], rituximab should be offered as initial treatment to MN patients at moderate or high risk, whereas the Ponticelli’s regimen should be considered for those at very high risk of progression (1B). Rituximab is also recommended as a therapeutic option at first recurrence or when anti PLA2R antibodies did not become negative after six months. Either the rheumatological schedule (1 g given two weeks apart) or the classical weekly 375 mg/m^2^ dose for four weeks can be chosen as a dose.

Several case series and retrospective studies suggest the efficacy and safety of rituximab also in MCD and FSGS [30,31,32,33]. In particular, rituximab seems particularly useful in the subset of patients who are steroid- or calcineurin- dependent. In 2014, the Rituximab in Nephrotic Syndrome of Steroid-Dependent or Frequently Relapsing Minimal Change Disease or Focal Segmental Glomerulosclerosis (NEMO) Study [34] reported a decrease in the relapse rate in a cohort of 30 patients with steroid-dependent or frequently relapsing nephrotic syndrome in remission at the time of treatment with rituximab (a single 375 mg/m^2^ dose in most cases). Conversely, rituximab was found of little use in an Italian cohort of 8 FSGS patients who had biopsy-proven FSGS with contraindications to corticosteroids or conventional immunosuppression [35]. These opposing efficacy data could be partially due to an insufficient characterization of the different FSGS forms (primary, secondary or genetic). In this regard, segmental versus diffuse foot process effacement or disease onset with either full-blown nephrotic syndrome or sub-nephrotic proteinuria could be of help in identifying the patients on whom insisting with immunosuppression therapy [36]. Recently, a meta-analysis of 16 observational studies on either FSGS or MCD seemed to support the efficacy of rituximab in these nephropathies; as expected, patients with MCD achieved remission in higher percentage in comparison to those with FSGS [37].

In recent years, some randomised, clinical trials provided further evidence on the efficacy and safety of rituximab in children with steroid-resistant or steroid-dependent nephrotic syndrome compared to placebo [38], steroids [39,40] or calcineurin inhibitors [40,41].

As for MN, new KDIGO guidelines suggest a wider use of rituximab in patients with MCD or FSGS [22]. In MCD, rituximab can be used as initial therapy in those patients who have contraindications to glucocorticoids or better in those who are frequent relapsing or steroid dependent.

## 4. Pitfalls and Open Issues of Rituximab Therapy in Nephrotic Syndrome

Even if rituximab is an effective treatment for MN, a significant percentage of patients do not achieve a satisfactory decrease of proteinuria. Several reasons have been hypothesised to explain treatment failure in MN, in general or more specifically in those receiving rituximab.

First of all, available evidence for rituximab use comes from trials testing different treatment schedule and doses. Probably the higher the dose, the better the results, as testified by the finding of the MENTOR trial in MN [24]. However, lower doses have not been tested properly by long-term randomized clinical trials. Indeed, in the GEMRITUX trial [23], the primary endpoint was evaluated only at six months and the patients were treated with two doses of 375 mg/m^2^ each.

Patients with high titres of baseline anti-PLAR2 antibody are more resistant to treatment [42,43,44]. In this subset of patients, rituximab efficacy can be improved by a second course of treatment [45]. Similarly, high titres of anti-SOD2 and anti-α-enolase antibodies at diagnosis were associated with poor clinical outcome, with patients having combined positivity showing the poorest one [8]. PLA2R1 epitope spreading has been reported as another marker of poor outcome and treatment failure (even if this has been recently contradicted) [8,46]. In the future, the serological classification could better characterise the patients and possibly personalise treatment schedules accordingly [47].

The development of neutralising anti-rituximab antibodies has also been described and is probably underestimated in clinical practice [48].

Also in MCD and FSGS, several open questions are pending on rituximab doses and schedules. Some studies showed remission of proteinuria with lower total rituximab doses, questioning the benefit of additional dosing [49,50,51]. However, in children with SSNS, a higher rate of relapse and a shorter time to first relapse have been described in those who received lower doses of rituximab [52,53,54]. These data may support that rituximab is characterised by a dose-dependent efficacy.

## 5. The Nephrotic Syndrome and Rituximab Plasma Levels

The administration schedules of rituximab for nephrotic syndrome were borrowed from those in use for non-Hodgkin’s lymphoma (NHL) or autoimmune diseases. However, in nephrotic patients, the pharmacokinetics profile of rituximab can be different in comparison to other treatment indications, as the degree of proteinuria may have a significant effect on rituximab levels. Indeed, patients with MN have significantly lower rituximab serum concentrations in comparison to subjects with rheumatoid arthritis [55] or myasthenia gravis with no proteinuria [48], while receiving the same rituximab schedule. Interestingly, rituximab concentrations were higher at retreatments, possibly as a consequence of an improvement of the nephrotic syndrome. In patients with MN and nephrotic syndrome, rituximab also has a much shorter half-life in comparison to that observed in subjects with myasthenia gravis [48], NHL or rheumatoid arthritis [56].

It has been hypothesised that rituximab could be lost in the urine at a higher extent following a loss in the selectivity of the glomerular membrane (as it is the case in many forms of nephrotic syndrome) [33,57,58]; the urinary loss of rituximab in nephrotic syndrome may cause insufficient drug exposure and shorter serum drug half-life. In severe cases, rituximab half-life could be extremely reduced (less than one day), also because of pleural and peritoneal fluid loss [57,59]. It is then possible that poor results in children with steroid-resistant nephrotic syndrome may partially be due to insufficient rituximab plasma levels [51,53].

Consistently, MN patients treated with higher rituximab doses have higher serum levels, lower CD19+ counts and lower level of anti-PLA2R1 antibodies at six months [60]. However, serum rituximab levels in the 15 days following rituximab infusion did not differ significantly between responders and non-responders [55]. However, low residual rituximab levels at month 3 were shown to be correlated with poor B-cell depletion at months 3 and 6, with high anti-PLA2R1 titre at months 3, 6, and 12, and with proteinuria at months 3, 6 and 12 [48].

## 6. The Correlation between CD19+ Depletion and Treatment Response

CD19+ B-cell depletion occurs fast (within few hours) and almost all patients achieve it, even after receiving small doses when the infusion of the drug is stopped prematurely due to side effects (personal experience). At three months, circulating CD19+ B cells start to recover, but at six months most patients still have counts below the reference range. Afterwards, CD19+ lymphocytes progressively re-emerge into the circulation from month six to the end of the first year, usually returning to pre-treatment levels within twelve months (even if some patients maintain reduced CD19+ levels for much longer periods). However, the repopulation rate is extremely variable, depending on rituximab doses, underlying disease, previous immunosuppression and individual factors [61,62]. For instance, the repopulation rate seems to be longer in patients with systemic vasculitis [63] compared to those affected by rheumatoid arthritis [64]. On the contrary, it seems to be shorter in patients with systemic lupus erythematosus [65].

Some authors reported that the achievement and duration of clinical remission correlate well with CD19+ cell in MCD-FSGS [66,67] and MN [68,69]. Conversely, relapses are often preceded by B-cell repopulation. This has been well described in children with steroid-dependent (SDNS) and frequent-relapsing (FRNS) nephrotic syndrome [53,70,71,72]. Accordingly, a “CD19-targeted therapy” has been proposed to avoid unnecessary additional infusions or prevent relapses when needed. In this respect, Ramachandran et al. [73] reported their clinical experience on 109 MN patients who were treated with three different dosing regimens (375 mg/m^2^ weekly for four weeks, 1 g on days 0 and 15 and CD19-guided rituximab therapy. No difference was found in the response rate to the different rituximab schedules, but the patients receiving the CD19-guided scheduled regimen needed a lower rituximab dose and possibly had a better safety profile. Good results were obtained also in 53 adults with MCD/FSGS with SDNS, FRNS and steroid-resistant nephrotic syndrome who received CD19-targeted rituximab while in remission [74]. Most of the patients maintained a sustained remission over a median follow-up of 36 months allowing the withdrawal of calcineurin inhibitors and steroids in nearly 80% of the cases. The mean total dose of rituximab at one year was 788.7 ± 128.1 mg.

Unfortunately, this approach is limited by the fact that the correlation between CD19+ levels and the response to treatment is weak in many instances. Indeed, despite complete peripheral CD19+ B-cell depletion, not all the patients have a clinical response to rituximab [23,75,76,77]. On the contrary, some patients who do not achieve B-cell depletion have an improvement in proteinuria [51,78,79]. Relapses during a full B-cell depletion have been also described [33,80,81]. Finally, some treated patients maintain long-term clinical remission even after total B-cell recovery [53,70,72,82,83].

Even if B-cell depletion is easy to achieve with low-dose rituximab, insufficient dosing may delay remission [56] or expose the patient to a higher rate of relapse and to a shorter time to the first relapse compared to patients receiving higher doses [66,82].

Altogether, CD19+ count alone may not be sufficient alone for driving optimal dosing and administration schedule of rituximab in relation to the single patient needs and clinical course.

In recent years, the concept is emerging that other cellular biomarkers could be more accurate than the mere CD19+ count. In particular, memory B cells could escape rituximab-mediated depletion or repopulate earlier, thus favouring relapses. Memory B cells expressing the immunoglobulin-G (IgG)-switched phenotype are irreversibly committed to plasma cell differentiation and are considered as the reservoir of potential pathogenic autoantibodies, as opposed to IgD/IgM memory and naïve B cells. Long-lived plasmacells lose the CD20 surface antigen, are refractory to conventional immunosuppression, including rituximab and hard to eliminate [84].

Moreover, traditional flow cytometric (FCM) analyses may lack of sufficient sensitivity to detect very low CD19+ count level that could burst or maintain active immunological activity.

## 7. High-Sensitivity Flow Cytometric Cell Counting for Immune Monitoring of Anti-CD20 Therapies

In the last decade, the measurement of B cell levels and their functional subsets has been greatly influenced by the technical evolution of FCM techniques. In particular, the availability of high-sensitivity, multi-colour, FCM techniques has made possible the precise detection and count of extremely low levels of B cells and B cell subsets in the peripheral blood of patients receiving anti-CD20 MoAbs [14,85,86].

A standardised and easily applicable FCM immune monitoring protocol (ISCCA Protocol) has been recently developed to accomplish this task [85]. A number of technical requirements are needed to ensure a precise and reproducible analysis: (i) a high-resolution technique including at least one million leucocytes (the more, the better) to allow the acquisition of a sufficient number of B cells and their small subsets which can be greatly reduced under treatment; (ii) a multi-colour FCM technique using 8 fluorescence channels and 10 cellular markers [86]; (iii) a timing scheme that includes serial measurements at pre-defined timepoints during the patients’ follow-up to adjust the therapy protocol on an individual basis. An example of a simplified analysis protocol is depicted in Figure 1.

A simplified gating strategy in a healthy subject with the acquisition of about 1,126,000 leukocyte events is represented. In each diagram the axes labels indicate the respective cell surface markers and fluorochrome types, as described in detail in the reference by Gatti et al., 2020 [86]. Lymphocytes are captured and divided into T, B, and NK cells. B cells are cleaned from contaminants and divided into CD19+ CD27− naïve and CD19+ CD27+ memory cells. Memory B cells are further dissected into surface IgG-switched and IgM memory cells. T cells are divided into the CD4+ and CD8+ subsets. The table reports the relevant cell event numbers and the percentages of each cell subsets with reference to the respective parent population, to be included in the clinical report.

The ISCCA Protocol includes the simultaneous enumeration of T-cell subsets and NK lymphocytes. This is of importance since some anti-CD20 MoAbs are dependent from the availability of functional Fc-gamma receptors on NK cells [87]. Moreover, a relatively preserved T cell and Treg repertoire is a prerequisite to prevent excess immunosuppression after anti-CD20 treatments and possibly to cooperate in the cytotoxic mechanisms [87,88].

Another important feature of the ISCCA Protocol is the further subsetting of CD19+ CD27+ memory B cells into surface IgG+ (committed memory Ig-switched) and IgM+ (relatively immature memory B) cells. This provides a deeper insight of the functional status of the memory B cell response, since the IgG switch represents a point of no return in B cell activation. The latter point has proved valuable in dissecting patients with rheumatoid arthritis with different responses to therapy [14,86] and could be applied in other immune-mediated disorders, including autoimmune renal diseases (Figure 2).

Marked differences in the relative proportions of IgG-switched and IgM memory B cells can be present in different patients and can be associated with various degrees of disease activity and response to therapy. This indicates that the evaluation of the mere percentage of memory B cells is not informative enough in clinical immune monitoring. Four representative cases of patients with autoimmune disorders are depicted (in columns A, B, C and D, respectively), with opposite IgG/IgM distributions, despite similar levels of total memory B cells.

## 8. High-Sensitivity Flow Cytometric Cell Counting in Clinical Practice

A wider use of quantitative monitoring protocols may provide clinicians with additional information to evaluate the clinical response and better calibrate immunosuppression in the single patient, especially when nephrotic proteinuria acts as a confounding factor.

In most of the patients, an operational B cell ‘disappearance’ (i.e., <0.01/μL by high-resolution FCM) is obtained after the first rituximab dose, and may last up to 8–10 months. Besides, the repopulation of as few as 0.5 B cells/μL can be considered as an indicator that the effect of rituximab is wearing off. Looking at B cell subsets, sustained depletion of CD27+ memory B cells and the reappearance of naïve B cells are more precise indicators of clinical response [79,89,90]. In particular, the repopulation of an unbalanced B cell repertoire, with an excess of CD19+ CD27+ memory B cells and a lower proportion of CD19+ CD27- naïve B cells are considered as an indicator of impending relapse [14].

The kinetics of B-cell repopulation was analysed in one patient with MN by means of using fluorescence-activated cell sorter analysis [91]. Rituximab rapidly induced complete disappearance of CD19+ B cells, plasmablasts, and memory B cells. Despite persistence of CD19+ depletion, plasmablasts and memory B cells re-emerged early before naive B cells.

More recently, it has become possible to directly delete surface IgG and IgM expression and thus to identify IgG-switched CD19+ CD27+ cells. This is a particular subset of memory B cells that represents the earliest population potentially endowed with the pathogenic autoantibody memory [92].

## 9. New Anti-CD20 Targeted Therapies for Nephrotic Syndrome

A great deal of evidence has been accumulated over the last decade on the use of newly developed anti-B cell treatments in autoimmune disorders, such as ocrelizumab, ofatumumab and obinutuzumab, each endowed with different potency, a different balance of the respective cytotoxic mechanisms, and half-lives [93,94]. Despite belonging to the same pharmaceutical class and hitting the same cell target than rituximab, the new anti-CD20 MoAbs should be considered as different drugs in many respects [94]. Given their different mechanism of action on B-cell depletion and being fully humanised, they possibly overcome resistance or immunogenicity towards rituximab.

Ofatumumab recognizes an epitope encompassing both the small and large extracellular loops of the CD20 antigen. It has been shown to be effective in rituximab-resistant nephrotic syndrome [95]. Moreover, it brings a faster depletion of anti-PLA2R antibody when associated with double-filtration plasmapheresis [96]. Recently, Ravani et al. [97] showed similar efficacy and safety of ofatumumab in comparison to rituximab in 140 children and young adults (aged 2–24 years) with nephrotic syndrome maintained in remission with prednisone and calcineurin inhibitors. Unfortunately, the marketing authorization of ofatumumab was withdrawn in the European Union for commercial reasons.

Obinutuzumab is the first recombinant IgG1 type-II anti-CD20 monoclonal antibody with Fc engineered to optimize the interaction with cellular Fc-γReceptor III-A. This change makes obinutuzumab more efficient in antibody-dependent cellular cytotoxicity (ADCC), in opsonic phagocytosis, and in inducing direct apoptosis of the cell target, as compared with rituximab [93,94]. Obinutuzumab also recognizes a CD20 epitope different from that of rituximab and was proven effective in rituximab-resistant haematological malignancies [94]. Anecdotally, it was found effective in the case of membranoproliferative glomerulonephritis secondary to CLL [98]. Recently, the Mayo Clinic Group described successful treatment with obinutuzumab of three cases of treatment-resistant MN [99].

Ocrelizumab is a humanized B cell depleting antibody, approved for treatment of multiple sclerosis (MS) [100]. As compared to rituximab, it displays an increased ADCC and a reduced complement-dependent cytotoxicity, while showing a longer half-life. Recently, a patient with MS and PLA2R-associated MN was successfully treated with this drug [101].

## 10. Future Developments

Anti-CD20 therapies have become an important treatment strategy of several glomerulonephritis. With increasing use, a wide patient- and disease-related variability of the clinical response to treatment has emerged. Although the use of high-dose rituximab protocols may obtain good clinical responses, one must pay a high price in terms of side effects and associated morbidity (especially in aged patients). On the contrary, some patients would need much lower doses to obtain remission and avoid relapses and are treated unnecessarily with high doses and/or repeated treatments.

Anti-CD20 therapies are a unique opportunity to directly measure the net immunosuppressive effect on the relevant cell target, to monitor the occurrence of escape or resistance phenomena, and to evaluate the timing and the cell composition of the post-treatment B cell recovery. In this context, the use of new objective biomarkers could help in tailoring the therapeutic regimens on each patient’s clinical features (especially to avoid over-immunosuppression).

However, recent studies have shown that following anti-CD20 treatment other lymphocyte functional mechanisms may further play a role, thus making the interpretation of the biological findings more complex and sometimes controversial. Evidence is becoming available that the spleen can be a sanctuary where rituximab-resistant memory B cells with a peculiar phenotype and autoreactive plasma cells may settle [102]. Atypical ‘double-negative’ memory B cells expressing the CD19+ CD27− sIgD− sIgG+ phenotype have been demonstrated and hypostasised to be the real precursors of pathogenic autoantibodies [103]. Double-negative memory B cells are also partly overlapped with CD19+ CD21− CD11c+ Tbet+ Age-related B cells that accumulate during inflammaging. Rituximab also affects T cell and myeloid cell functions, while repopulating naïve B cells appear with a more activated phenotype, as compared to their pre-rituximab baseline counterparts [104].

The more in-depth immune system cells are studied, the wider the heterogeneity of functional cell subset and response or regulatory mechanisms is evidenced, thus necessitating a continuous upgrade of our knowledge and diagnostic panoply.

## Figures and Tables

**Figure 1 jcm-10-05847-f001:**
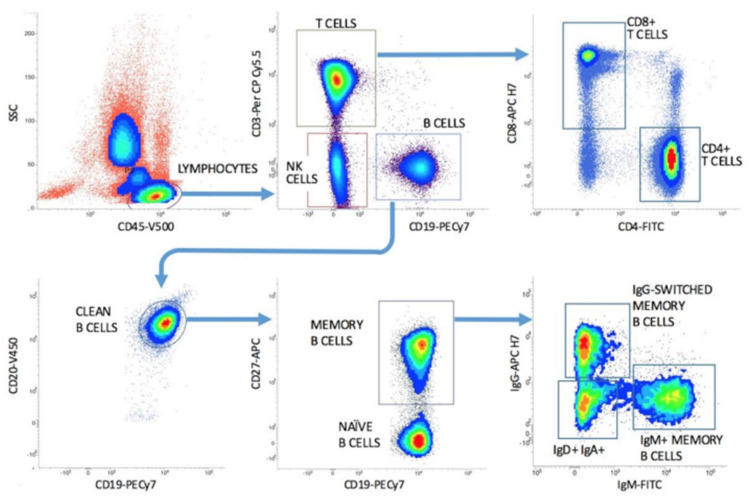
The ISCCA Protocol for the high-resolution analysis of functional B cell subsets.

**Figure 2 jcm-10-05847-f002:**
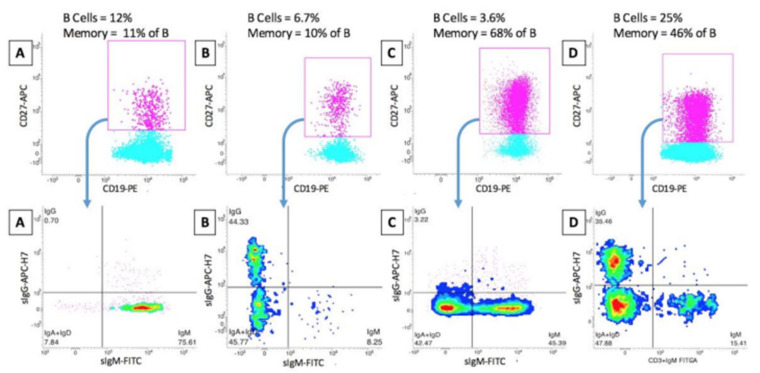
The functional heterogeneity of CD19+ CD27+ memory B.

**Table 1 jcm-10-05847-t001:** Main efficacy and safety data of the GEMRITUX, MENTOR, and STARMEN trials in membranous nephropathy.

CLINICAL TRIAL	GEMRITUX [23]	MENTOR [24]	STARMEN [27]	RI-CYCLO [28]
Country	France	The United States and Canada	Spain, France and the Netherlands	Italy and Switzerland
Publication Year	2017	2019	2020	2021
Randomized patients (*n*)	75	130	86	74
Inclusion criteria	Proteinuria ≥3.5 g/day or a urinary protein-to-creatinine ratio ≥3500 mg/g, and serum albumin <30 g/L for at least 6 months with maximal dose of RAS blockade eGFR >30 mL/min/1.73 m^2^	Proteinuria >5 g/day on two samples Decline of <50% in proteinuria despite NIAT for at least 3 months before randomization eGFR or creatinine clearance >40 mL/min/1.73 m^2^	Proteinuria > 4 g/day after 6 months of observation Hypoalbuminemia (≤3.5 g/dL) eGFR ≥45 mL/min/1.73 m^2^	Proteinuria >3.5 g/day on three 24-h urine collections (once a week for 3 weeks), on RAS blockade eGFR ≥30 mL/min/1.73 m^2^
Rituximab group	375 mg/m^2^ on days 1 and 8 in association with NIAT	1000 mg on days 1 and 15 A second course administered if proteinuria was reduced by at least 25% at 6 months but no complete remission Continuing NIAT	Oral tacrolimus (0.05 mg/Kg/day), to reach target blood levels of 5–7 ng/mL, for six months. At day 180, rituximab (1 g) and tacrolimus dosage was reduced by 25% per month, with complete withdrawal at the end of month 9	1000 mg on days 1 and 15 Continuing NIAT
Control group	NIAT	Cyclosporine with a starting dose of 3.5 mg/kg/day. Target trough blood levels of 125 to 175 ng/mL Tapering according to remission status at 6 months (total treatment length from 6 to 14 months) Continuing NIAT	1 g × 3 methylprednisolone at months 1, 3 and 5, then 0.5 mg/Kg/day orally from day 4 to 30). At months 2, 4 and 6, oral cyclophosphamide adjusted for age and renal function (1.0–2.0 mg/Kg/day for 30 days)	1 g × 3 methylprednisolone at months 1, 3 and 5, then 0.5 mg/Kg/day orally from day 4 to 30). At months 2, 4 and 6, oral cyclophosphamide (2.0 mg/Kg/day for 30 days) Continuing NIAT
Baseline characteristics:				
- Mean Age (years)	56	Around 52	55.7	55
- Body weight (kg)	76	Around 90	78.5	76
- Protein-to-creatinine ratio, mg/g	7363.2 (4702.9–9735.0)	NA	NA	NA
- Proteinuria (g/day)	NA	8.9 (median)	7.4 (median)	6
- eGFR (mL/min/1.73 m^2^)	68.6	NA	79.8	84
- CrCl - (mL/min/1.73 m^2^)	NA	Around 86	NA	NA
- Positive Anti-PLAR2 (%)	73.3	73.8	77%	66%
Primary endpoint	Complete or partial remission of proteinuria at 6 months	Complete or partial remission of proteinuria at 24 months	Complete or partial remission of proteinuria at 24 months	Complete or partial remission of proteinuria at 12 months
Primary outcome	*n* = 13 (35.1%; 95% [95% CI, 19.7 to 50.5) in the rituximab group and *n* = 8 (21.1%; 95% CI, 8.1 to 34.0) in the NIAT group; *p* = 0.21	*n* = 39 (60%) in the rituximab group and *n* = 13 (20%) in the cyclosporine group (risk difference, 40 percentage points; 95% CI, 25 to 55; *p* < 0.001 for both non-inferiority and superiority).	*n* = 36 (83.7%) in the corticosteroid-cyclophosphamide group and *n* = 25 (58.1%) in the tacrolimus-rituximab group (RR 1.44; 95% CI 1.08 to 1.92)	*n* = 23 (62%) in the rituximab group and *n* = 27 (73%) receiving the cyclic regimen (OR, 0.61; 95% CI, 0.23 to 1.63).
Anti–PLA2R-Ab trend	At 6 months deletion in13 of 26 (50%) in the rituximab group and 3 out of 25 (12%) in NIAT group (*p* = 0.004)	In the subgroup achieving partial or complete remission higher reduction of anti PLAR2 ab titre for rituximab group in comparison to cyclosporine at all-time points	Significant decrease in both treatment groups. A higher proportion of anti-PLA2R-positive patients achieved immunological response at 3 and 6 months in the corticosteroid-cyclophosphamide group (77% and 92%, respectively), as compared to the tacrolimus-rituximab group (45% and 70%, respectively)	Anti-PLA2R levels decreased in both arms during follow-up, more rapidly in the rituximab arm.
Safety	Eight (21%) serious adverse events in each group	Serious adverse events in 11 patients (17%) in the rituximab group and in 20 (31%) in the cyclosporine group (*p* = 0.06)	More adverse events and more adverse events per patient in the corticosteroids-cyclophosphamide group than in the tacrolimus-rituximab group (*p* = 0.04).	Serious adverse events occurred in 19% of patients receiving rituximab and in 14% receiving SOFIJA the cyclic regimen.

Legend: GEMRITUX, Evaluate Rituximab Treatment for Idiopathic Membranous Nephropathy; MENTOR, Evaluate Rituximab Treatment for Idiopathic Membranous Nephropathy; RAS, Renin Angiotensin System; NIAT, non-immunosuppressive antiproteinuric treatment; eGFR, estimated glomerular filtration rate; CrCl, creatinine clearance; Anti–PLA2R-Ab, antiphospholipase A2 receptor antibody; RI-CYCLO, Rituximab versus Steroids and Cyclophosphamide in the Treatment of Idiopathic Membranous Nephropathy; STARMEN, Sequential Treatment with Tacrolimus and Rituximab versus Alternating Corticosteroids and Cyclophosphamide in PMN study.

## Data Availability

Not applicable.

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
