# Peer review of "Rituximab Therapy for Adults with Nephrotic Syndromes: Standard Schedules or B Cell-Targeted Therapy?"

_jcm, 2021, doi:10.3390/jcm10245847_

Round 1
Reviewer 1 Report
Authors propose a review on rituximab therapy in adults with idiopathic NS, both MCD/FSGS and MN and discuss the presently challenging views on dose regimen and immunological monitoring.
I have minor comments
1/ About references
P6 The authors mention « Several cas series and restrospective studies suggest the efficacy of RTX also in MCD and FSGS (30-31-32) ». Despite the review concerning adult nephrotic syndrome, I would recommend to refer and/or discuss one or more randomized control trials that have demonstrated the efficacy/safety of rituximab in pediatric FR/SDNS :
Iijima K, Sako M, Nozu K, Mori R, Tuchida M, Kamei K, et al. Rituximab for childhood-onset, complicated, frequently relapsing nephrotic syndrome or steroid-dependent nephritic syndrome: a multicentre, double-blind, randomised, placebo controlled trial. Lancet 2014;384 (9950):1273-81
Ravani_P, Rossi_R, Bonanni_A, Quinn_RR, Sica_F, Bodria_M, et al. Rituximab in children with steroid-dependent nephritic syndrome: a multicenter, open-label, noninferiority, randomized controlled trial. Journal of the American Society of Nephrology 2015;26(9):2259-66.
Ahn_YH, Kim_SH, Han_KH, Choi_HJ, Cho_H, Lee_JW, et al. Efficacy and safety of rituximab in childhood-onset, difficult-to-treat nephrotic syndrome: a multicenter open-label trial in Korea. Medicine 2018;97(46):
Basu B, Sander A, Roy B, Preussler S, Barua S, Mahapatra TKS, Schaefer F. Efficacy of Rituximab vs Tacrolimus in Pediatric Corticosteroid-Dependent Nephrotic Syndrome: A Randomized Clinical Trial. JAMA Pediatr. 2018 Aug 1;172(8):757-764
Human or Chimeric Monoclonal Anti-CD20 Antibodies for Children with Nephrotic Syndrome: A Superiority Randomized Trial. Ravani P, Colucci M, Bruschi M, Vivarelli M, Cioni M, DiDonato A, Cravedi P, Lugani F, Antonini F, Prunotto M, Emma F, Angeletti A, Ghiggeri GM. J Am Soc Nephrol. 2021 Oct;32(10):2652-2663
P7 Concerning the discussion on the optimal dosage/regimen of RTX in children SSNS, references 47-48 could be updates with more recent / wider studies , for example :
Chan E, Webb H, Yu E, Ghiggeri GM, Kemper MJ, Lap-Tak Ma A and al, Both the rituximab dose and maintenance immunosuppression in steroid-dependent/frequently-relapsing nephrotic syndrome have important effects on outcomes. 2020 Feb;97(2):393-401.
Hogan J, Dossier C, Kwon T, Macher MA, Maisin A, Couderc A, et al. Effect of different rituximab regimens on B cell depletion and time to relapse in children with steroid-dependent nephrotic syndrome. Pediatr Nephrol. 2019 Feb;34(2):253-259
Maxted A, Dalrymple R, Chisholm D, McColl J, Tse Y , Christian M, Ben C Reynolds and al, Low-dose rituximab is no less effective for nephrotic syndrome measured by 12-month outcome. Pediatr Nephrol 2019 May;34(5):855-863
Minor concerns
P6 RY-CICLO should be replaced by RI-CYCLO (3 times)
P9 Titles for figures are missing.
Author Response
Thank you for your suggestions.
As required, we added extra references on more randomized control trials that have demonstrated the efficacy/safety of rituximab in pediatric FR/SDNS.
We also update references 47-48 and corrected the misspelled ry-ciclo
We added the missing titles
kind regards
Lucia Del Vecchio
Reviewer 2 Report
I would like to congratulate the authors on this manuscript, one of the best I have read/ reviewed in the last year!. I have few suggestions to it:
- in the abstract, would consider adding a statement at the end to mention the main points you will be discussing in this manuscript.
- in table 1 and page 7, would correct the reference (appears as Error bookmark).
- Consider adding references to support your statements in page 7 about the time lines of CD 19 recovery, as these can vary between different studies.
- In page 8, statement: “Altogether, CD 19+ count alone..” I would consider revising this to say something like: “it may not be sufficient alone” or “ we believe…” . I fully agre with your statement and reason is that we published a review manuscript about Ritux in NS and some reviewers were against such a strong statement.
Good luck!
Author Response
thank you for you nice comments
we corrected the manuscript according to the suggestions:
-we added a statement in the abstract to mention the main points we discuss in the manuscript
-we did not spot in our word files the reference mistakes in table 1 and pg
we added some reference and extra information on CD19 trend over time and on its variability
- we modified the statement at pg 8